# The impact of school absence on mental health in children and young people: Analysis of an English national birth cohort

**Russell M. Viner**[1]*, **Anna Pearce**[2], **Steven Hope**[1,3]

**1** Population, Policy and Practice Research Department, UCL Great Ormond St. Institute of Child Health, London, United Kingdom, **2** MRC/CSO Social and Public Health Sciences Unit, School of Health and Wellbeing, University of Glasgow, Glasgow, United Kingdom, **3** School of Public Health, Imperial College London, London, United Kingdom

* r.viner@ucl.ac.uk

## Abstract

### Background

Given concerns the role of school closures in increasing mental health problems after the Covid-19 pandemic, we used pre-pandemic data to undertake a causal epidemiological analysis of the associations of school absence with later mental health problems.

### Methods

Longitudinal data from the Millennium Cohort Study were collected pre-pandemic at ages 7 (in 2008), 11 (2012) and 14 years (2015), securely linked with English routine educational data on school absence in the 2 years preceding each cohort wave. We constructed marginal structural models for mental health problems (as outcome) and quartiles of absence (exposure), taking account of baseline and time-varying confounding.

### Results

Those in the highest quartile of absence had odds ratios (OR) for experiencing later mental health problems of 2.216 (1.629, 3.014) at age 7 (n = 6383), and in lagged models OR of 1.508 (1.072, 2.122) at age 11 (n = 6102) and 1.903 (1.234, 2.934) at 14 years (n = 5616). Persistent absence (>10% of school year) was associated with OR for later mental health problems of 2.00 (1.56, 2.57) at 7 years, and in lagged models OR of 2.26 (1.62, 3.14) at 11 years and 2.00 (1.27, 3.16) at 14 years.

**Data availability statement:** The data for the Millennium Cohort Study are held in a public depository, the UK Data Archive. All data used in this paper are publically available on special application through the UK Data Service Secure Lab (https://ukdataservice.ac.uk/help/secure-lab/). University College London, UCL Institute of Education, Centre for Longitudinal Studies, Department for Education. (2024). Millennium Cohort Study: Linked Education Administrative Datasets (National Pupil Database), England: Secure Access. [data collection]. 3rd Edition. UK Data Service. SN: 8481, DOI: http://doi.org/10.5255/UKDA-SN-8481-3 University of London. Institute of Education. Centre for Longitudinal Studies. (2017). Millennium Cohort Study: Third Survey, 2006. [data collection]. 7th Edition. UK Data Service. SN: 5795, http://doi.org/10.5255/UKDA-SN-5795-4 University of London. Institute of Education. Centre for Longitudinal Studies. (2017). Millennium Cohort Study: Fourth Survey, 2008. [data collection]. 7th Edition. UK Data Service. SN: 6411, http://doi.org/10.5255/UKDA-SN-6411-7 University of London. Institute of Education. Centre for Longitudinal Studies. (2017). Millennium Cohort Study: Fifth Survey, 2012. [data collection]. 4th Edition. UK Data Service. SN: 7464, http://doi.org/10.5255/UKDA-SN-7464-4 University of London. Institute of Education. Centre for Longitudinal Studies. (2018). Millennium Cohort Study: Sixth Survey, 2015. [data collection]. 3rd Edition. UK Data Service. SN: 8156, http://doi.org/10.5255/UKDA-SN-8156-3.

**Funding:** The author(s) received no specific funding for this work.

**Competing interests:** The authors have declared that no competing interests exist.

## Conclusions

School absence above the second quartile doubled the odds of later mental health problems in both primary and secondary school children in pre-pandemic data. Our findings support there being a strong and potentially causal association between absence from school and later mental health problems, and suggest that absence from school is harmful for CYP's mental health.

## Introduction

High levels of both absence from school [1] and poor child and adolescent mental health [2] in many wealthy countries in the years since the COVID-19 pandemic have focused attention on relationships between school non-attendance and student mental health and wellbeing. There is a large literature showing that poor school attendance is associated with mental health problems, with mechanisms postulated to include social isolation from peer groups, isolation from supportive adults including teachers and pastoral staff, learning loss and removal from the safety-net of essential child support services that schools enable [3–8]. However systematic reviews [5,7,8] have concluded the quality of studies in this area is poor, with the great majority of studies being cross-sectional. This is problematic because there is likely a reciprocal relationship between school absence and mental health, with mental health problems causing children to miss school and school absence worsening mental health and wellbeing [9].

Any discussion of post-pandemic mental health amongst children and young people (CYP) must take into account the COVID-19 pandemic, which witnessed a rise in child mental health problems in many countries [2,10]. In England, probable mental health problems rose from 1 in 9 immediately pre-pandemic to 1 in 6 in 2020 and 2021 [2]. Post-pandemic, rates appear stable amongst 7–16 year olds although 16–17 year olds experienced a further increase to 1 in 4 in 2022 [11]. Whilst enforced absenteeism due to school closures is distinct from absence while schools are open, this form of absence has been argued to be a key factor in these increases through social isolation and learning loss alongside other factors including pandemic anxiety, bereavement, economic and family strains and post-covid effects [12–14].

Education policy requires unbiased longitudinal estimates of the impact of school absence on later mental health problems in CYP. These issues remain highly pertinent for England and many other countries 4–5 years post-pandemic as school absence rates remain stubbornly high. Further, the COVID-19 experience suggests that in future pandemics there may again be high levels of student absenteeism while schools are open, reflecting high parental anxiety. Pre-pandemic data offers the opportunity to study the impacts of school absence without having to take into account pandemic impacts such as school closures. We therefore used a nationally representative UK birth cohort with linked routine education administrative data to estimate the effects of school absence on mental health using marginal structural models and controlling for a wide set of plausible confounders. We hypothesized that higher absence increased the risk of later mental health problems.

## Methods

The MCS is an ongoing population-based study representative of all children born in the UK in 2000–2001. The MCS oversampled children from economically disadvantaged areas and areas with a high proportion of ethnic minorities. Participant recruitment and selection is described elsewhere [15]. Routine educational data on school absence and educational attainments from the National Pupil Database (NPD) [16] were linked to MCS participants in England only, with data only available for those attending public schools.

We examined the association of mental health measured in surveys at ages 7, 11 and 14 years [17–19] with school attendance in the 2 years preceding each survey. We used a directed acyclic graph (DAG) to define the temporal sequence related to school absence (as exposure) and mental health problems (as outcome) and the hypothesized confounding structure of other important variables (Fig 1). Our choice of hypothesized confounding variables was guided by the literature noted above.

### Outcomes

Mental health problems at each of age 7, 11 and 14 years were assessed by parent-completed Strengths and Difficulties Questionnaire (SDQ), extensively validated as a measure of mental health in UK populations [20]. Scores above published thresholds (SDQ 'high scorer', defined as a score>=17) at each age were used as a proxy for probable mental health disorder [21].

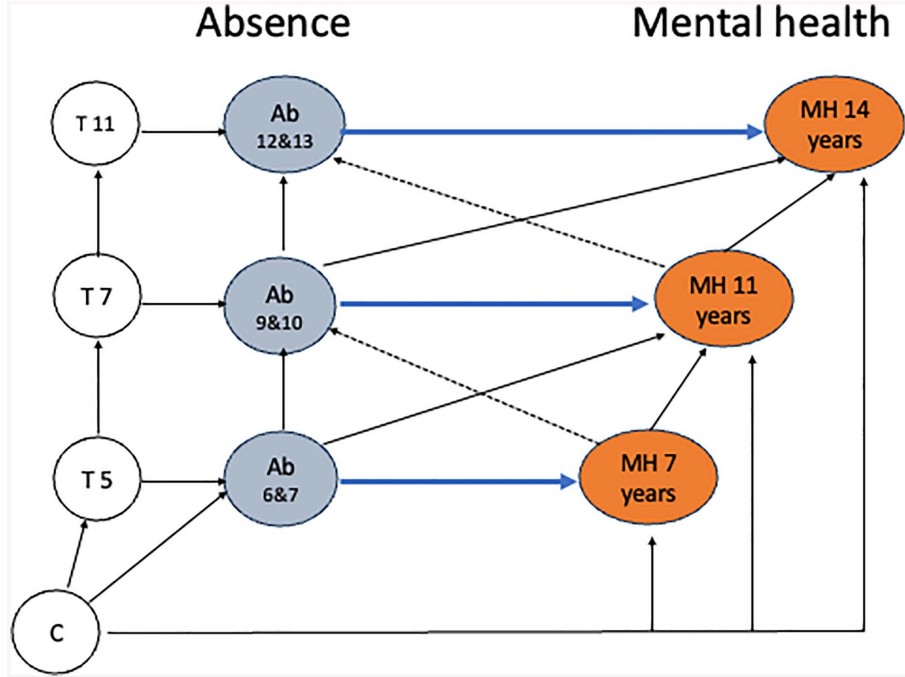

**Fig 1. Causal diagram of hypothesized pathways between school absence (Ab) and child mental health (MH) at ages 7, 11 and 14 years.** Figure shows a directed acyclic graph (DAG) of hypothesized pathways between school absence (Ab) and child mental health (MH) at ages 7, 11 and 14 years. Ab: school absence in 2 years prior to MH measurement (i.e., 6&7 years, 9&10 years, 12&13 years). MH: high scorer on SDQ at 7, 11 and 14 years. C: baseline confounders: measured before children started school: sex, ethnicity and deprivation. Note that C baseline confounders were included in each of the models, including for Ab9&10 and Ab12&13 however only a single arrow is shown in Fig 1 for simplicity and as C confounders were only included once in lagged models. T: time varying confounders: maternal mental health, long-term condition, previous educational attainments, engagement with school, special educational needs and bullying, area-level deprivation, prior SDQ.

## Exposures

The NPD [16] provided data for each child on the number of half-day school sessions for which a student was authorized to be absent and sessions absent without authorization and the possible number of sessions for each child. The Department for Education (DfE), which collected the absence data, defines "authorised absence" as absence with permission from a teacher or other authorised representative of the schools, e.g., for illness [22]. Given these analyses aimed to understand the impact of total school absence on mental health, we combined both authorised and unauthorised absences.

To ensure exposures were prior to the outcome, we defined our years of interest for school absence as being the 2 years prior to each MCS survey at which SDQ data were available; for the age 7 survey (in 2008) we used absence data for 2007 and 2008 as no absence data were available for 2006); absence data were used for 2009 and 2010 for the age 11 survey (2012) and 2012 and 2013 and for the age 14 survey (2014−15). We averaged sessions absent for each child over the relevant two years and then calculated the proportion of possible sessions that were absent.

The association of absence with mental health problems is unlikely to be linear, and we chose to categorize absence in quartiles as our primary analysis, using the lowest (fewest sessions absent) quartile as the reference group. In supplementary analyses we a) examined authorized absence as a predictor of later mental health; and b) examined associations using total absence as a continuous logged variable and as a binary variable defining persistent absence as > 10% of possible sessions, consistent with the definition of 'persistent absence' used by the DfE [22].

## Confounding structure

We included the following potential confounding variables (Fig 1):

Baseline confounding: Potential confounding factors measured before children started school included sex, ethnicity and residential area-level deprivation [3] (index of multiple deprivation (IMD)).

Time-varying confounding: The following variables, measured in the sweep preceding exposure measurement were adjusted for: maternal mental health (Kessler Psychological Distress scale); maternal report of presence of a long-term condition in the child [23], school experience [24] including engagement with school, bullying victimization and special or additional educational needs (SEND); and previous educational attainments obtained from linked NPD data (national data collected at age 5 years (Key Stage 1) and at age 10–11 years (Key Stage 2)). Where a required confounding variable was not available at a wave, the relevant variable from the preceding wave was substituted to ensure temporal precedence was maintained.

## Analysis

We constructed marginal structural models (MSM) weighted using inverse probability weights (IPW) calculated from a minimally-sufficient confounding structure identified from the DAG. We first we estimated odds ratios (OR), with 95% confidence intervals (CIs) at 7, 11 and 14 years, in separate models, with the exposure being absence in the previous 2 years and the IPW taking account of baseline confounding and time-varying confounders relevant to each age. The IPWs also included mental health problems (SDQ high scorer) measured at the previous wave.

We then ran 'cross-lagged' MSMs at 11 and 14 years which took account of previous associations between absence, SDQ and time-varying confounding (e.g., for 11 year taking into account relationships and variables at 7 years; for 14 years taking into variables at 7 and 11 years), following Hope 2018 [25]. For these cross-lagged MSM we used a combined IPW calculated by multiplication of IPW for the current analysis with that for the previous analyses. We confined our analyses to complete cases as missingness was relatively low for all included variables. We did not use population weighting methods provided for MCS analyses as these are not applicable for analyses across multiple waves as we have undertaken. All analyses were undertaken in Stata 16 (StataCorp, College Station, TX) within the secure environment of the UK Data Service SecureLab.

Ethics: No ethical approval necessary for these secondary analyses of anonymized data. Data owner permissions were obtained through the UK Data Service.

## Results

Characteristics of the sample for exposure (school absence) and outcome (mental health) are shown in Table 1 with characteristics of other confounding variables used to construct the IPW shown in Table 2.

The sample for these analyses consists of those children with data on both school absence and mental health disorder at each age: n = 7659 at 7 years, n = 6689 at 11 years, n = 5838 at 14 years. Proportions of children with probable mental health problem (SDQ high scorer) were 8.0% at age 7 years, 9.1% at 11 years and 10.2% at 14 years. The mean proportion of time that children were absent from school was 5.3% in the two years before age 7, 3.6% in the 2 years before age 11 and 4.2% in the 2 years before age 14. The majority of absence was authorized at all ages, with unauthorized absence accounting for only 0.49% to 0.57% of possible school sessions across ages. The sample for these analyses was broadly representative of the child population of England, being 50.7% male and 80.8% white at age 7, with similar proportions at 11 and 14 years. Consistent with the MCS recruitment strategy, 15% of children were in the most deprived decile at age 7, with 8–12% in all other deciles.

**Table 1. Characteristics of children in terms of school absence in preceding 2 years and mental health disorder at 7, 11 and 14 years.**

| | | | 7 years (2008) | 11 years (2012) | 14 years (2015) |
|---|---|---|---|---|---|
| A. ABSENCE | | | Absence 2007−08 | Absence 2010−11 | Absence 2012−13 |
| Total number of sessions | | Mean | 608.33 (607.59, 609.06) n = 7659 | 607.92 (607.19, 608.64) n = 6689 | 684.69 (683.78, 685.59) n = 5838 |
| | | Median | 616 | 614 | 694 |
| Absence (authorized) | Annual sessions absent | Mean | 29.22 (28.68, 29.76) n = 7659 | 24.56 (24.05, 25.07) 6689 | 24.61 (24.02, 25.20) n = 5838 |
| | | Median | 24 | 20 | 19 |
| | % absent of total sessions | Mean | 4.83 (4.74, 4.92) n = 7659 | 4.05 (3.97, 4..13) n = 6689 | 3.61 (3.51, 3.69) n = 5838 |
| | | Median | | 3.25 | 2.73 |
| Absence (unauthorized) | Annual sessions absent | Mean | 2.92 (2.71, 3.11)) n = 7659 | 3.10 (2.90, 3.30) n = 6689 | 3.83 (3.56, 4.10) n = 5838 |
| | | Median | 0 | 0 | 0 |
| | % absent of total sessions | Mean | 0.49 (0.46, 0.53)) n = 7659 | 0.52 (0.49, 0.56) n = 6689 | 0.57 (0.53, 0.61) n = 5838 |
| | | Median | 0 | 0 | 0 |
| Total Absence (authorized and unauthorized) | Annual sessions absent | Mean | 32.14 (31.52, 32.75)) n = 7659 | 27.09 (26.66, 28.24) n = 6689 | 28.44 (27.74, 29.15) n = 5838 |
| | | Median | 25 | 22 | 21 |
| | % absent of total sessions | Mean | 5.31 (5.22, 5.42)) n = 7659 | 4.57 (4.47, 4.67) n = 6689 | 4.18 (4.07, 4.28) n = 5838 |
| | | Median | 4.18 | 3.57 | – |
| B. MENTAL HEALTH | | | (n = 7659) | (n = 6689) | (n = 5838) |
| Probable mental health disorder | SDQ high scorer | % (95% CI) | 8.04 (7.44, 8.67) | 9.10 (8.43, 9.82) | 10.23 (9.46, 11.03) |

Notes: Table shows the characteristics of children in these analyses, i.e., A. the distribution of school absence (authorized, unauthorized and total absence) at each age, characterized by the mean and median; and B. the proportions and 95% confidence intervals (95% CI) with probable mental health problems (defined as being high scorer on the Strengths and Difficulties Questionnaire (SDQ)). Data disclosure rules operating in the SecureLab meant that inter-quartile ranges (IQR) and minimum and maximum scores could not be disclosed due to small sample sizes. Where median is not reported, the sample size for the median value also lay below disclosure rules.

**Table 2. Cohort member characteristics for confounding variables.**

| | | 7 years (2008) | 11 years (2012) | 14 years (2015) |
|---|---|---|---|---|
| | | % (n) | % (n) | % (n) |
| Sex | Male | 50.65 (3870) | 50.07 (3349) | 50.27 (3210) |
| | Female | 49.35 (3771) | 49.93 (3340) | 49.73 (3175) |
| Ethnicity | White | 80.81 (6189) | 81.24 (5434) | 79.14 (4620) |
| | Mixed ethnicity | 1.24 (95) | 1.17 (78) | 1.18 (69) |
| | Indian | 3.45 (264) | 3.23 (216) | 3.80 (222) |
| | Pakistani or Bangladeshi | 8.16 (625) | 8.43 (564) | 9.66 (564) |
| | Black | 4.47 (342) | 3.93 (263) | 4.01 (234) |
| | Other | 1.87 (143) | 1.99 (133) | 2.21 (129) |
| Index of multiple deprivation (IMD) decile | | % (n) | % (n) | % (n) |
| | 1: Most deprived decile | 15.0 (1,100) | 14.29 (956) | 13.2 (843) |
| | 2 | 12.11 (888) | 11.75 (786) | 11.14 (711) |
| | 3 | 11.19 (821) | 10.85 (726) | 10.73 (685) |
| | 4 | 9.48 (695) | 9.06 (606) | 9.21 (588) |
| | 5 | 9.86 (723) | 9.73 (651) | 10.05 (642) |
| | 6 | 8.43 (618) | 9.15 (612) | 9.43 (602) |
| | 7 | 8.41 (617) | 8.87 (593) | 9.19 (587) |
| | 8 | 8.22 (603) | 8.4 (562) | 8.54 (545) |
| | 9 | 8.41 (617) | 8.64 (578) | 9.08 (580) |
| | 10: Least deprived decile | 8.89 (652) | 9.25 (619) | 9.43 (602) |
| Enjoys school | Always | 72.04 (5230) | 64.15 (4291) | 64.51 (3766) |
| | Usually | 24.16 (1754) | 29.54 (1976) | 29.27 (1709) |
| | Sometimes | 3.33 (242) | 5.34 (357) | 5.28 (308) |
| | Not at all | 0.47 (34) | 0.84 (56) | 0.77 (45) |
| Bullied at school | Never | 65.24 (4997) | 65.67 (4393) | 65.60 (3824) |
| | Once or twice | 27.30 (2091) | 26.94 (1802) | 27.05 (1579) |
| | Several times | 5.54 (424) | 5.47 (366) | 5.53 (323) |
| | Many times | 1.67 (128) | 1.58 (106) | 1.54 (90) |
| Long-term condition | No | 80.18 (5859) | 81.25 (5427) | 85.96 (4569) |
| | Yes | 19.82 (1448) | 18.75 (1252) | 14.04 (746) |
| Child has special needs | No | – | 91.90 (6147) | 92.03 (5373) |
| | Yes | – | 7.95 (532) | 7.74 (452) |
| KS1 reading & writing score level | Level 3 or above | 12.48 (886) | | |
| | Level 2b or above | 47.96 (3404) | | |
| | Level 2 or above | 20.78 (1475) | | |
| | Level 1 or below | 18.78 (1333) | | |
| KS1 achieved Maths level | Level 3 or above | 23.47 (1666) | | |
| | Level 2b or above | 53.47 (3795) | | |
| | Level 2 or above | 14.48 (1028) | | |
| | Level 1 or below | 8.58 (609) | | |
| | | Continuous | continuous | continuous |
| Key Stage 2 point score | Mean (sd) | – | 28.79 (3.37) n = 6654 | 28.80 (4.46) n = 5825 |
| | Median | – | 27 n = 2923 | 27 n = 2568 |
| Maternal mental health: Kessler score | Mean (sd) & n | 3.29 (3.81) n = 6938 | 3.25 (3.89) n = 6389 | 3.28 (3.92) n = 5515 |
| | Median | 2 | 2 | 2 |

Notes: Table shows the characteristics of the sample in terms of the confounding variables that were used in constructing the inverse probability weighting (IPW). For categorical variables, proportion (%) and sample size (n) are shown. For continuous variables, the mean, median and sample size (n) are shown. Ethnicity was collapsed for analyses into the categories shown.

Table 3 shows odds ratios (OR) and 95% confidence intervals (CI) for risk of probable psychological disorder associated with quartiles of total school absence in the previous 2 years, adjusted for confounding factors and previous measurements of the outcome in unlagged and lagged models (that took account of previous relationships between absence and mental health). Final model samples were 6383 at 7 years (83.3% of the analytic sample at 7 years), 6102 at 11 years (91.2% of the analytic sample) and 5616 at 14 years (96.2% of analytic sample).

In unlagged models, there was a dose response relationship between absence and mental health problems across all ages, with those in the highest quartiles of absence at around twice as likely (or more) to experience mental health problems as those in the lowest quartile. For example, odds ratios (OR) for experiencing mental health problems (highest quartile of absence compared with the lowest) was 2.216 (1.629, 3.014) at age 7, 1.919 (1.368, 2.693) at age 11 and 2.690 (2.045, 3.539) at 14 years.

In lagged models, the dose response relationships were still seen, but these were weaker than in the individual models. For example, the increased risk of mental health problems in the highest quartile of absences was (OR 1.5 (1.071, 2.122) at 11 years and 1.903 (1.234, 2.934) at 14 years. In the cross-lagged models, there no longer appeared to be an elevated risk in the second absence quartile.

Supplementary analyses Table 4 shows models for absence as a continuous (log transformed) variable and for persistent absence (absence >10%). In the continuous analyses, the risk of later mental health problems increased with duration of school absence at 7 years but not at 11 or 14 years in either unlagged or lagged analyses. Persistent absence

**Table 3. Odds ratios for probable psychological disorder at 7, 11 and 14 years associated with quartiles of preceding school absence in unlagged and lagged mediation models.**

| | 7 years (re absence aged 6–7 year) Model n = 6383 | | | 11 years (re absence aged 10-11y) Model n = 6102 | | | 14 years (re absence aged 12-13y) Model n = 5616 | | |
|---|---|---|---|---|---|---|---|---|---|
| *Unlagged Models* | % absence in quartiles | OR (95% CI) | p-value | % absence in quartiles | OR (95% CI) | p-value | % absence in quartiles | OR (95% CI) | p-value |
| Absence quartiles: Ref: Lowest quartile | <2.2% | 1 | | <1.9% | 1 | | <1.5% | 1 | |
| 2nd quartile | 2.3-4.2% | 1.163 (0.825, 1.638) | 0.388 | 2.0-3.6% | 1.201 (0.866, 1.666) | 0.272 | 1.5-3% | 1.272 (0.937, 1.725) | 0.122 |
| 3rd quartile | 4.3-7.1% | 1.421 (1.031, 1.958) | 0.032 | 3.7-6.2% | 1.433 (1.041, 1.971) | 0.027 | 3.1-5.7% | 1.935 (1.459, 2.566) | <0.0001 |
| Highest quartile | 7.2% plus | 2.216 (1.629, 3.014) | <0.0001 | 6.2% plus | 1.919 (1.368, 2.693) | <0.0001 | 5.8% plus | 2.690 (2.045, 3.539) | <0.0001 |
| *Lagged models for 11 and 14 years* | | | | | Model n = 5517 | | | Model n = 4597 | |
| Absence quartiles: Ref: Lowest quartile | | | | <1.9% | 1 | | <1.5% | 1 | |
| 2nd quartile | | | | 2.0-3.6% | 1.185 (0.857, 1.638) | 0.304 | 1.5-3% | 1.038 (0.696, 1.549) | 0.856 |
| 3rd quartile | | | | 3.7-6.2% | 1.249 (0.898, 1.736) | 0.186 | 3.1-5.7% | 1.759 (1.172, 2.641) | 0.006 |
| Highest quartile | | | | 6.2% plus | 1.508 (1.072, 2.122) | 0.018 | 5.8% plus | 1.903 (1.234, 2.934) | 0.004 |

Notes: Table shows odds ratios (OR) and 95% confidence intervals (CI) for risk of probable psychological disorder associated with total school absence in the previous 2 years, adjusted for confounding factors as shown in Table 2. Note that absence quartiles were derived separately for each of the 2 year periods prior to age 7, 11 and 14 years. Models are shown unlagged and lagged. The unlagged models only use inverse probability weights (IPW) derived for that analysis (separate IPW were derived for models for 7, 11 and 14 years). Lagged models were then run for 11 and 14 years taking account of previous absence and previous associations between mental health and absence, i.e., the model for 11 years used the product of the IPW for 7 and 11 years, and the model for 14 years used the product of the IPW for 7, 11 and 14 years.

**Table 4. Supplementary analyses: odds ratios for probable psychological disorder at 7, 11 and 14 years associated with preceding school absence as a continuous or binary variable in unlagged and lagged mediation models.**

| | 7 years (re absence aged 6–7 year) Model n=6383 | | 11 years (re absence aged 10-11y) Model n=6102 | | 14 years (re absence aged 12-13y) Model n=5616 | |
|---|---|---|---|---|---|---|
| **Unlagged Models** | OR (95% CI) | p-value | OR (95% CI) | p-value | OR (95% CI) | p-value |
| School absence (continuous; anti-logged) | 1.373 (1.136, 1.658) | 0.001 | 1.030 (0.935, 1.133) | 0.552 | 1.153 (0.969, 1.372) | 0.109 |
| Persistent absence (Absence >10%) | 2.005 (1.564, 2.568) | <0.0001 | 2.98 (2.166, 4.093) | <0.0001 | 2.370 (1.811, 3.100) | <0.0001 |
| **Lagged models for 11 and 14 years** | | | Model n=5517 | | Model n=4597 | |
| School absence (continuous; anti-logged) | | | 1.055 (0.957, 1.163) | 0.281 | 1.074 (0.935, 1.233) | 0.312 |
| Persistent absence (>10%) | | | 2.255 (1.620, 3.138) | <0.0001 | 1.999 (1.267, 3.155) | 0.003 |

Table shows odds ratios (OR) and 95% confidence intervals (CI) for risk of probable psychological disorder associated with total school absence in the previous 2 years, adjusted for confounding factors. Models are shown for total absence first as a continuous variable and second as a dichotomous variable (highly absent, defined as absence >10%), and third in quartiles of absence. Models are shown unlagged and lagged. The unlagged models only use inverse probability weights (IPW) derived for that analysis (separate IPW were derived for models for 7, 11 and 14 years). Lagged models were then run for 11 and 14 years taking account of previous absence and previous associations between mental health and absence, i.e., the model for 11 years used both IPW for 7 and 11 years, and the model for 14 years used IPW for 7, 11 and 14 years.

was found in 11.7% in 2007−08, 8.2% in 2009−10 and 6.8% in 2012−13. Being persistently absent was associated with OR for later mental health problems of 2.00 (1.56, 2.57) at 7 years and in lagged models OR were 2.26 (1.62, 3.14) at 11 years and 2.00 (1.27, 3.16) at 14 years. A sensitivity analysis examining authorized absence (see Table 5) showed similar findings.

## Discussion

We found that levels of absence above the second quartile of the distribution of sessions missed were associated with greater odds of mental health problems 1–2 years later. This is a relatively low threshold for school absence, and suggests that the upper half of the population distribution of school sessions missed were at risk of mental health problems related to absence. Estimates for lagged models for 11 and 14 years were smaller than in unlagged models, suggesting that failing to account for confounding histories likely results in upwards-biased estimates. We found that the increased risk for mental health problems was apparent from only approximately 4% total absence amongst 7 and 11 year olds, with an approximately 40% increase in odds of mental health problems, although in lagged models for age 11 risk was only significantly increased from around 6% absence. For 14 year olds, increased risk was apparent from approximately 3% total absence, but with larger effect sizes, with an 80% increase in odds of mental health problems from approximately 3% absence and a 90% increase from approximately 6% absence. The risk of mental health problems among those experiencing persistent absence compared to those who did not, was approximately double at all ages.

A number of studies have shown links between school absence and mental health [3–8], with persistent or prolonged absenteeism most strongly associated with these problems [26]. This is true for emotional or internalizing disorders, behavioural or externalizing problems [27] and for those with neurodevelopmental problems [3]. Both absences that are authorized by school and those that are not (unauthorized absence) have been associated with mental health problems [23], with relationships stronger for secondary compared to primary school children [23]. Data from the English National Child and Adolescent Mental Health survey in early 2022 found that those with a mental health disorder were nearly twice as likely to have persistent absence as those without problems [11]. However the majority of studies are cross-sectional.

Most research posits mental health problems as a result of school absence, with poor peer relationships [28,29] and isolation [13] as mechanisms. However, there is some longitudinal evidence that relationships may also run from poor mental health to later absence [5], truancy and school drop-out [30]. Pathways linking absence with mental health

**Table 5. Odds ratios for probable psychological disorder at 7, 11 and 14 years associated with preceding *authorized* school absence in unlagged and lagged mediation models.**

| | 7 years | | | | | 11 years | | | | | 14 years | | | | |
|---|---|---|---|---|---|---|---|---|---|---|---|---|---|---|---|
| **Unlagged models** | | *OR* | *p-value* | *95% CI upper* | *95%CI lower* | | *OR* | *p-value* | *95% CI upper* | *95%CI lower* | | *OR* | *p-value* | *95% CI* | *CI* |
| | N = 6383 | | | | | N = 6102 | | | | | N = 5616 | | | | |
| Absence quartiles: Ref: Lowest quartile | <2.1% | | | | | <1.7% | | | | | <1.4% | | | | |
| 2nd quartile | 2.1-3.8% | 1.158 | 0.397 | 0.825 | 1.625 | 1.7-3.2% | 1.292 | 0.122 | 0.933 | 1.789 | 1.4-2.7% | 1.171 | 0.296 | 0.870 | 1.576 |
| 3rd quartile | 3.9-6.4% | 1.318 | 0.091 | 0.957 | 1.814 | 3.3−5.5% | 1.412 | 0.034 | 1.027 | 1.941 | 2.8-4.9% | 1.732 | 0.000 | 1.313 | 2.284 |
| Highest quartile | 6.5%+ | 2.124 | <0.0001 | 1.571 | 2.872 | 5.6%+ | 1.965 | <0.0001 | 1.411 | 2.736 | 5.0%+ | 2.385 | 0.000 | 1.828 | 3.112 |
| **Lagged models, i.e., 11 and 14y accounting for previous absence and previous associations between mental health and absence** | | | | | | | | | | | | | | | |
| | | | | | | N = 5517 | | | | | N = 4597 | | | | |
| Absence quartiles: Ref: Lowest quartile | | | | | | <1.7% | | | | | <1.4% | | | | |
| 2nd quartile | | | | | | 1.7-3.2% | 1.312 | 0.099 | 0.950 | 1.811 | 1.4-2.7% | 1.073 | 0.725 | 0.724 | 1.592 |
| 3rd quartile | | | | | | 3.3−5.5% | 1.256 | 0.167 | 0.909 | 1.737 | 2.8-4.9% | 1.423 | 0.072 | 0.969 | 2.092 |
| Highest quartile | | | | | | 5.6%+ | 1.545 | 0.010 | 1.112 | 2.147 | 5.0%+ | 1.551 | 0.038 | 1.025 | 2.349 |

Notes: Table shows odds ratios (OR) and 95% confidence intervals (CI) for risk of probable psychological disorder associated with authorized school absence in the previous 2 years, adjusted for confounding factors as shown in Table 3. Models are shown for 2 variables for total absence; first as a continuous variable and second in quartiles of unauthorized absence. Note that quartiles were derived separately for each of 7, 11 and 14 years. Models are shown unlagged and lagged. The unlagged models only use inverse probability weights (IPW) derived for that analysis (separate IPW were derived for models for 7, 11 and 14 years). Lagged models were then run for 11 and 14 years taking account of previous absence and previous associations between mental health and absence, i.e., the model for 11 years used both IPW for 7 and 11 years, and the model for 14 years used IPW for 7, 11 and 14 years. All N are unweighted. Sample sizes are shown by model.

problems are likely to be complex. First, there may be reciprocity between the two, so that absence may lead to poor mental health due to isolation [13], but poor mental health may increase the chances of children missing or avoiding school. Second, there may be confounding by educational attainments and success, as those with poorer mental health are more likely to have poor attainments [31], but poor attainments are associated with absence [4]. Third, wider social determinants may confound relationships between absence and mental health, and they both may be influenced by a range of shared risk factors. Both absence and poor mental health may reflect common underlying social determinants such as deprivation and family factors [29,32], although few studies have examined these factors [5]. Fourth, key confounders of the relationship between absence and mental health such as bullying are likely to change over time during childhood and adolescent, however studies have thus far failed to account for time-varying confounding. Studies have also largely failed to recognise that previous absence is one of the strongest predictors of school absence [33].

## What this study adds

Ours is the first study in this area to use causal inference methods in nationally representative longitudinal data, taking account of temporality, time-varying confounding and earlier associations between absence and mental health. Standard statistical methods to adjust for confounding that is influenced by previous exposure can lead to biased estimates [34]. First, conditioning on a confounder that is affected by previous exposure removes part of the effect of interest. Second, if there are unmeasured factors associated with both school absence and mental health, conditioning or adjusting for a

confounding variable (as is done in standard regression) potentially induces further confounding through unmeasured pathways from exposure to outcome via these unmeasured variables, known as 'collider bias' [34]. We used weighted marginal structural models (MSM) [34,35] to account for the complex hypothesised pathways between exposure (school absence), outcome (mental health), baseline confounding and time-varying confounding. MSMs are an effective method of generating unbiased potentially causal estimates in the presence of time-varying confounding in social epidemiology [36].

We identified a potentially causal association between school absence and later mental health problems in primary and secondary school children, and identified low thresholds (e.g., 3–4% total absence) above which risk of mental health problems increased.

## Limitations of this study

Our data are subject to a number of limitations. While we were able to adjust for a range of confounders, these will not have been measured perfectly and it is possible that there are other important confounders that we have not adjusted for. Our analyses were based upon our DAG, which included relationships between mental health and absence at previous waves and thus accounted for potential bi-directional relationships between mental health and absence. However we cannot exclude unobserved residual confounding explaining some of our findings.

Not all MCS participants in England had linked educational data, likely reflecting loss during the linkage process and lack of consent for linkage. It is likely these data are missing at random as non-linkage is unlikely to be related to any of the variables under study here. There was minor excess attrition in the sample for cohort members from the most deprived deciles and from some ethnic groups; whilst this is unlikely to impact the substantive relationship between school absence and mental health problems, the association of deprivation with both absence and mental health problems suggest our findings may under-estimate its size.

For analytic parsimony we only examined overall mental health problems using the SDQ total difficulties scale, rather than examine associations of absence with different types of mental health problems, such as self-reported anxiety or depression.

We used pre-pandemic data to examine these associations, as this inherently excluded a range of pandemic-related confounding factors and allowed us to more cleanly study school absence as an exposure. However, we recognise that prepandemic absence from school is an imperfect analogy for enforced absence due to school closures during the COVID-19 pandemic and therefore the associations observed in this study might have changed during and since the pandemic.

## Policy implications and conclusions

Our findings support there being a strong and potentially causal association between absence from school and later mental health problems in primary and secondary school CYP. Absence for more than 3–4% of total school-time significantly increased the odds of later mental health problems, with greater impact in adolescence. This threshold is notably lower than the 10% threshold used to define persistent absence in England. Whilst higher levels of post-pandemic absence across the school population mean that quartiles of school absence will be different to those identified here, our findings suggests that mental health harms related to absence are likely at lower levels of school absence.

Our use of prepandemic data within a causal mediation framework allows us to be confident that absence from school is harmful for CYP's mental health separate to any impacts from other pandemic stressors.

Estimates suggest that pupils in England lost 61 days of schooling on average between March 2020 and April 2021, approximately 32% of a school year [37], far higher than the thresholds used in this study. These findings will be important for policymakers, given the persisting levels of absence post-pandemic seen in countries such as England and in relation to decisions regarding school closure in any future pandemics. There is a need for further study of the association

between absence and mental health in the post-pandemic period. Our data suggest that persisting high levels of school absence post-pandemic are contributing to rising levels of mental health problems amongst young people, and support conceptualisation of school absence as a public health problem. Our findings therefore support action on reducing school absence as a potentially useful part of efforts to improve children and young people's mental health.

## Author contributions

**Conceptualization:** Russell M. Viner.

**Data curation:** Russell M. Viner.

**Formal analysis:** Russell M. Viner.

**Methodology:** Russell M. Viner, Anna Pearce, Steven Hope.

**Writing – original draft:** Russell M. Viner.

**Writing – review & editing:** Russell M. Viner, Anna Pearce, Steven Hope.

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
