## [Decision Letter · Decision Letter 0]

18 Jan 2025

PONE-D-24-53350The impact of school absence on mental health in children and young people: analysis of an English national birth cohortPLOS ONE

Dear Dr. Viner,

Thank you for submitting your manuscript to PLOS ONE. After careful consideration, we feel that it has merit but does not fully meet PLOS ONE’s publication criteria as it currently stands. Therefore, we invite you to submit a revised version of the manuscript that addresses the points raised during the review process.

Please ensure that your revised manuscript addresses all reviewer comments comprehensively; however, I would like to draw your particular attention to the following three points:

**Disentangle school closures and absenteeism:** The manuscript conflates these concepts, which have distinct implications, and we recommend clarifying their differences.**Endogeneity concerns in the causal relationship:** Despite controlling for maternal mental health, the bidirectional relationship between absenteeism and mental health raises concerns that should be acknowledged.**Mechanisms linking absenteeism to mental health:** The manuscript could benefit from a brief discussion on how absenteeism disrupts key benefits provided by schools, such as structure, social interaction, and essential services.

We look forward to receiving your revised manuscript.

Kind regards,

Leonard Moulin

Academic Editor

PLOS ONE

Journal Requirements:

Reviewers' comments:

Reviewer's Responses to Questions

**Comments to the Author**

1. Is the manuscript technically sound, and do the data support the conclusions?

Reviewer #1: Yes

Reviewer #2: Yes

2. Has the statistical analysis been performed appropriately and rigorously? 

Reviewer #1: Yes

Reviewer #2: Yes

3. Have the authors made all data underlying the findings in their manuscript fully available?

Reviewer #1: Yes

Reviewer #2: No

4. Is the manuscript presented in an intelligible fashion and written in standard English?

Reviewer #1: Yes

Reviewer #2: Yes

5. Review Comments to the Author

Reviewer #1: The authors equate school closures with school absences. However, these are distinct phenomena with different implications for students, educators, and communities. School closures, such as caused by the Covid-19 pandemic, result in the inability of all students to attend their educational institution. In contrast, school absences generally refer to the nonattendance of individual students or smaller groups due to personal reasons, such as illness, family responsibilities, or truancy.

By conflating these two terms, the authors risk oversimplifying the challenges and misaligning proposed solutions. For example, strategies effective in addressing individual absenteeism may be inadequate for managing the systemic disruptions of school closures. Similarly, the long-term academic, social, and emotional consequences differ, requiring nuanced understanding and action. It is essential to disentangle these concepts to create precise and effective interventions that cater to the unique demands of each situation.

The authors rely on pre-pandemic data to make inferences about the post-pandemic world, but this approach has significant limitations. The Covid-19 pandemic has drastically altered the educational landscape, including shifts in teaching methods, technological reliance, student engagement, and systemic inequalities. Pre-pandemic data, while valuable for understanding historical trends, may fail to capture the profound and lasting changes brought about by the pandemic. Therefore, I would avoid the framing around the pandemic and focus on the relation ship between school absences and mental health generally. In addition, I would add this as limitation given that the relationship between school absences and mental health may have changed post-pandemic.

The authors fail to discuss why school absences should have a detrimental impact on adolescents' health and why the relationship between school attendance and health is bidirectional. This omission leaves a critical gap in their argument, as it does not address the underlying mechanisms that connect school attendance with health outcomes. The authors do not have to write about this extensively but should add a short discussion to the introduction, e.g., schools provide structured environments that promote routine, access to physical activity, social interaction, and critical services such as mental health counselling and nutritious meals. Absences disrupt these benefits, potentially leading to negative health outcomes such as increased social isolation, decreased physical activity, and higher stress levels due to academic setbacks or disconnection from peer networks.

The authors provide little information on handling missing data, a critical issue in longitudinal studies like the Millennium Cohort Study (MCS). Missing data, if unaddressed, can bias results and weaken validity. The authors should explicitly discuss the extent and patterns of missingness (e.g., MCAR, MAR, or MNAR) and adopt appropriate methods to address it.

Multiple imputation is recommended for handling missing data. This method generates several plausible datasets by filling in missing values based on observed patterns, then combines the results to produce robust estimates while accounting for imputation uncertainty. It is particularly suitable for complex datasets like the MCS.

The authors also use later waves of the MCS, which are subject to participant attrition. Systematic dropout can lead to nonrepresentative samples, introducing bias. To mitigate this, the authors should apply attrition weights to adjust for the likelihood of participants remaining in the study, ensuring the analysis more accurately reflects the original population.

The authors should elaborate on their use of the marginal structural model (MSM) and its advantages compared to standard regression techniques. My understanding is that MSMs are particularly suited for causal inference in longitudinal studies with time-varying exposures and confounders. They address issues where standard regression might fail, such as time-dependent confounding—when prior exposure influences both the outcome and subsequent confounders. This does not become clear. It may also be helpful to include equations to understand the estimation procedure.

While the authors’ use of a marginal structural model (MSM) is commendable for addressing time-varying confounding and bidirectionality, the analysis relies on several strong assumptions that warrant caution in interpreting causal claims. MSMs assume no unobserved confounding, meaning that all relevant confounders must be measured and included in the model. In practice, this is a strong assumption, as unmeasured variables could bias the results.

Additionally, the validity of the results depends on the correct specification of the parametric models used to calculate the weights. Misspecification can introduce substantial bias and undermine the causal interpretation of the findings.

Given these limitations, it is advisable to tone down causal language and frame the results as robust associations rather than definitive causal effects.

Reviewer #2: Thanks for this paper, it is good to see the linking to absence data to provide this kind of analysis. As mentioned in the article it serves as a rare more longitudinal and causal exploration of school attendance and mental health.

There are some minor and quick corrections to consider indicated below.

The only bigger picture thing to consider, which might only mean acknowledgement in introduction and discussion are the figures around school absence for the last few years which have increased dramatically. The introduction seems to focus on mental health difficulty increased and the potential effect of “absence” from school during lockdown. It therefore seems to miss the policy and practice emphasis on school attendance too and the potential implications of the findings if a risk factor for mental health has worsened since this data was collected.

Review the first sentence of Methods in abstract, it seems to be missing some words to make full sense.

The conclusions of the abstract don’t fit with the findings. Above the second quartile versus highest quartile and persistent absence in results summary. The final sentence of the abstract feels as though it is reaching given the timing of data here versus pandemic stressors.

Ought to be Department for Education throughout not “of”.

Page 4 missing end of sentence “However”

Given that the dataset is not available as used, could more be said about access to the NPD and the linking necessary, so this would be clear to other researchers interested in similar.

Could more be said to explain the decision to categorise into quartiles?

Could more be said to justify the selection of confounding variables.

Why are total session averages much higher in 2012-2013? (that could be explained in table 1 notes.

Table 1: Is 26.66 the correct total for 11 year olds?

The sample is described as broadly representative in terms of gender and ethnicity, is that also the case for amount of school absence?

Might a discussion point consider the alternative causal path from mental health difficulty to absence, the bi-directional relationship has been suggested in some research.

In the policy implications there seems to be a focus on days of schooling lost during lockdown. While this is relevant for the argument made in the paper, there ought to be acknowledgment that absence rates are higher in schools now compared to before pandemic.

I was surprised given the messages around 4% and 6% absence being a seemingly important threshold in this study’s findings, particularly as the profile of quartiles of absence is likely to be quite different now compared to pre-pandemic absence records, that an implication was not raised in terms of persistence absence and its cut-off. The findings seem to suggest that the arbitrary threshold of 10% of sessions missed might miss earlier identification of children who may be at risk of mental health difficulties if we were to use absence as an indicator of risk? Schools after all place emphasis on persistent absence as a key threshold to monitor.

6. PLOS authors have the option to publish the peer review history of their article (what does this mean? ). If published, this will include your full peer review and any attached files.

**Do you want your identity to be public for this peer review?** For information about this choice, including consent withdrawal, please see our Privacy Policy .

Reviewer #1: No

Reviewer #2: No

---

## [Author Response · Author response to Decision Letter 1]

1 Jun 2025

Response to reviewers

Dear Editor

Thank you for asking us to revise this paper. We respond point by point to the Editorial and reviewer comments below

Russell Viner on behalf of all authors

Editorial comments:

1. Disentangle school closures and absenteeism: The manuscript conflates these concepts, which have distinct implications, and we recommend clarifying their differences.

Response: We agree these are distinct and have distinct implications, although of course in both children are not in school. We have rewritten and restructured the introduction and discussion to emphasize that we are interested in school absence (i.e. absenteeism), but also to recognise that in the current post-pandemic generation these impacts can be difficult to separate from the forced absence due to school closures during the pandemic. Therefore we argue that examining pre-pandemic data can be helpful to ensure these impacts are distinct. We have removed any suggestion that these data might be helpful in understanding impacts of school closures, although we note that in future pandemics there may be very high rates of absenteeism while schools are open, due to parental anxiety, and that understanding such impacts upon mental health will be important for policy.

2. Endogeneity concerns in the causal relationship: Despite controlling for maternal mental health, the bidirectional relationship between absenteeism and mental health raises concerns that should be acknowledged.

Response: We undertook non-lagged and ‘lagged’ analyses specifically to account for prior associations between mental health and absence. Because of the temporal aspects of these variables, this did not require a directionality to the relationship i.e. it adjusted for the relationship at earlier periods regardless of directionality. We recognise that our DAG implied a directionality to this – although this is required of directed acyclic graphs. However our models through our DAG did allow us to account for previous bi-directional relationships between mental health and absence at earlier times. However we recognise in our revised limitations sections that residual confounding may remain.

3. Mechanisms linking absenteeism to mental health: The manuscript could benefit from a brief discussion on how absenteeism disrupts key benefits provided by schools, such as structure, social interaction, and essential services.

Response: We have included a brief discussion of this as suggested in the first paragraph of the revised introduction.

Review comments.

Reviewer #1

1. Re equating school absence with school closure

Response: see response to Editorial comment 1 above

2. Handling of missing data; use of attrition weights.

Response: Our analyses relate to a subset of the MCS for which linked educational data were available. This is a subset, albeit a large subset, of the overall dataset. It is therefore not a more standard MCS analysis. It is difficult to imagine on what basis one might multiply impute this linked dataset, as we note in our current limitations section that “Not all MCS participants in England had linked educational data, likely reflecting loss during the linkage process and lack of consent for linkage. It is likely these data are missing at random as non-linkage is unlikely to be related to any of the variables under study here.”

Similarly, attrition weights are not available for this linked dataset and we believe it would be incorrect to use weights for non-linked datasets. We note in our current limitations section that “There was minor excess attrition in the sample for cohort members from the most deprived deciles and from some ethnic groups; whilst this is unlikely to impact the substantive relationship between school absence and mental health problems, the association of deprivation with both absence and mental health problems suggest our findings may under-estimate its size.”

3. elaborate on their use of the marginal structural model (MSM) and its advantages compared to standard regression techniques

Response: We cite a number of papers in our methods section that outline the advances of MSM methods compared to standard regression techniques, and our Methods and What this study adds sections provide considerable detail on the advantages of these methods. We believe that additional detail is beyond the scope of this paper.

4. unobserved confounding

Response: we have strengthened our limitations section (as above) to better recognise the potential for unobserved confounding.

5. frame the results as robust associations rather than definitive causal effects

Response: We agree and have modified our language to conclude we have shown ‘strong and potentially causal’ associations using causal inference methods.

Reviewer 2.

1. issues about school absence and lockdown.

Response: see response to Editorial comment above

2. various typos and missing words

Response: Thank you, these have been corrected.

3. access to NPD

Response: We have added a note that “These data are available on special application through the UK Data Service Secure Lab (https://ukdataservice.ac.uk/help/secure-lab/).”

4. quartiles?

Response: We explain in our current methods section that we chose a categorical analysis as the association was very unlikely to be linear. This proved to be the case e.g. the medians were 0 as in Table 1. We chose quartiles as we wished to identify which sections of the child population were at risk, whether this was only those more extreme (i.e. highest quartile).

5. selection of confounding variables?

Response: We have revised our methods section to note “Our choice of hypothesized confounding variables was guided by the literature noted above.”

6. total absence sessions for 11 year olds?

Response: we had inverted the mean and the low confidence interval – thank you, this has been corrected.

7. representativeness in terms of absence?

Response: We do not have this information, yet this is a very large sample with linkage to routine administrative data, allowing the presumption of representativeness.

8. bidirectional causal pathways?

Response: see response to Editorial comments.

9. lockdown versus school absence?

Response: see response to Editorial comments.

10. Thresholds of absence – 4 and 6%

Response: we agree that the level of absence which appears to lead to mental health harms is lower than the UK definition of persistent absence at >10%. We have added the following to our conclusions: “This threshold is notably lower than the 10% threshold used to define persistent absence in England. Whilst higher levels of post-pandemic absence across the school population mean that quartiles of school absence will be different to those identified here, our findings suggests that mental health harms related to absence are likely at lower levels of school absence.”

---

## [Decision Letter · Decision Letter 1]

16 Jul 2025

PONE-D-24-53350R1The impact of school absence on mental health in children and young people: analysis of an English national birth cohortPLOS ONE

Dear Dr. Viner,

Thank you for submitting your manuscript to PLOS ONE. After careful consideration, we feel that it has merit but does not fully meet PLOS ONE’s publication criteria as it currently stands. Therefore, we invite you to submit a revised version of the manuscript that addresses the points raised during the review process.

One of the original reviewers has expressed satisfaction with the revisions made, while a new reviewer was invited and has provided a number of constructive comments and suggestions to be considered in the ongoing editorial process.

We look forward to receiving your revised manuscript.

Kind regards,

Leonard Moulin

Academic Editor

PLOS ONE

Journal Requirements:

Reviewers' comments:

Reviewer's Responses to Questions

**Comments to the Author**

1. If the authors have adequately addressed your comments raised in a previous round of review and you feel that this manuscript is now acceptable for publication, you may indicate that here to bypass the “Comments to the Author” section, enter your conflict of interest statement in the “Confidential to Editor” section, and submit your "Accept" recommendation.

Reviewer #2: All comments have been addressed

Reviewer #3: (No Response)

2. Is the manuscript technically sound, and do the data support the conclusions?

Reviewer #2: Yes

Reviewer #3: Yes

3. Has the statistical analysis been performed appropriately and rigorously? 

Reviewer #2: Yes

Reviewer #3: Yes

4. Have the authors made all data underlying the findings in their manuscript fully available?

Reviewer #2: No

Reviewer #3: Yes

5. Is the manuscript presented in an intelligible fashion and written in standard English?

Reviewer #2: Yes

Reviewer #3: Yes

6. Review Comments to the Author

Reviewer #2: Thanks for addressing the review points and the tracked changes on the manuscript.

I think there could have been a little more emphasis on school attendance as an increased concern now compared to during and prior to the pandemic, with some indication of how figures have worsened, particularly the near doubling of persistent absence. But the point around attendance as policy concern and caution around causation is now clear and this is important given the propensity for assuming poor mental health causes absence on less robust data and analysis than this.

Minor point for proofs, “Persistently absence” in the abstract should be “persistent absence”

Reviewer #3: The study “The impact of school absence on mental health in children and young people: analysis of an English national birth cohort“ uses inverse probability weighting and linked administrative and longitudinal survey data to estimate the effect of school absences on mental health. The study addresses an important topic, makes an important contribution to the literature, and uses high-quality data and a rigorous method. Nonetheless, there are a few points that need to be addressed before publication.

1. I suggest that the authors describe the method they used in the final sentence of the introduction, rather than stating that they “estimated the robust effect of school absence on mental health.” I recommend wording such as: “We use marginal structural models and control for a large set of plausible confounders to estimate the causal effect […]”.

2. The authors should describe the dataset in more detail. For example, it should be noted that the MCS oversampled children from areas with a high proportion of ethnic minorities and from economically disadvantaged areas. Additionally, it should be mentioned that the NPD only contains data on students attending public schools.

3. The authors should explicitly state that they did not use any survey weights, which is usually suggested for MCS data.

4. Please clarify exactly how many cases were excluded due to complete case analysis. Also, indicate how frequently missing values were replaced using data from the previous sweep.

5. Figure 1 does not include an arrow from C to Ab9&10 and Ab12&13. Was this omission intentional? Is C included in all models?

6. Why do the authors rely on parent-reported SDQ scores rather than self-reported SDQ scores? Are the results robust when using self-reported SDQ scores?

7. I do not understand why the authors present the results of the “unlagged model” in the abstract. Their DAG clearly shows that lagged absences and lagged mental health are confounders of the association between current absences and current mental health. The authors also acknowledge this in the discussion: “Estimates for lagged models for 11 and 14 years were smaller than in unlagged models, suggesting that failing to account for confounding histories likely results in upwards-biased estimates.” Therefore, if the authors aim to estimate the causal effect, they should focus exclusively on the results of the lagged model.

8. The authors need to provide a more detailed explanation of how they selected potential confounders. Their aim is to estimate the causal effect of absences on mental health, which can be achieved with marginal structural models only if all relevant confounders are included. Therefore, simply stating, “Our choice of hypothesized confounding variables was guided by the literature noted above,” is not sufficient. I recommend that the authors consider the risk factors for absences summarized in the meta-analysis by Gubbels et al. (2019), as many of these factors may also influence mental health.

9. In addition, I believe important confounders are missing, such as the socio-economic status of the family (which affects absences independently of area-level deprivation; Klein et al., 2020).

10. I do not fully understand the following sentences in the “limitations” section: “Not all MCS participants in England had linked educational data, likely reflecting loss during the linkage process and lack of consent for linkage. It is likely these data are missing at random as non-linkage is unlikely to be related to any of the variables under study here.”

First, it should be noted that the NPD does not contain information on students attending private schools. Second, do you mean “missing completely at random” (MCAR) or “missing at random” (MAR)? MCS-NPD linkage is related to demographics and socio-economic status, but conditional on these factors, it is not associated with internalizing and externalizing problems (Dräger et al. 2024, Appendix S1).

References

Dräger, J., Klein, M., & Sosu, E. M. (2024). Trajectories of school absences across compulsory schooling and their impact on children’s academic achievement: An analysis based on linked longitudinal survey and school administrative data. Plos one, 19(8), e0306716.

Gubbels, J., van der Put, C. E., & Assink, M. (2019). Risk factors for school absenteeism and dropout: A meta-analytic review. Journal of Youth and Adolescence, 48, 1637-1667.

Klein, M., Sosu, E. M., & Dare, S. (2020). Mapping inequalities in school attendance: The relationship between dimensions of socioeconomic status and forms of school absence. Children and Youth Services Review, 118, 105432.

7. PLOS authors have the option to publish the peer review history of their article (what does this mean? ). If published, this will include your full peer review and any attached files.

**Do you want your identity to be public for this peer review?** For information about this choice, including consent withdrawal, please see our Privacy Policy .

Reviewer #2: No

Reviewer #3: **Yes: ** Jascha Dräger

---

## [Author Response · Author response to Decision Letter 2]

18 Oct 2025

Please see attached cover letter

---

## [Editor Report · Decision Letter 2]

21 Oct 2025

The impact of school absence on mental health in children and young people: analysis of an English national birth cohort

PONE-D-24-53350R2

Dear Dr. Viner,

We’re pleased to inform you that your manuscript has been judged scientifically suitable for publication and will be formally accepted for publication once it meets all outstanding technical requirements.

Kind regards,

Leonard Moulin

Academic Editor

PLOS ONE
---

## [Editor Report · Acceptance letter]

PONE-D-24-53350R2

PLOS ONE

Dear Dr. Viner,

I'm pleased to inform you that your manuscript has been deemed suitable for publication in PLOS ONE. Congratulations! Your manuscript is now being handed over to our production team.

Kind regards,

on behalf of

Dr. Leonard Moulin

Academic Editor

PLOS ONE